# Cloud geometry from oxygen-A band observations through an aircraft side window

Tobias Zinner[1], Ulrich Schwarz[1], Tobias Kölling[1], Florian Ewald[1], Evelyn Jäkel[2], Bernhard Mayer[1], and Manfred Wendisch[2]

[1]Meteorologisches Institut, Ludwig-Maximilians-Universität, München, Germany
[2]Leipziger Institut für Meteorologie, Universität Leipzig, Germany

*Correspondence to:* Tobias Zinner (tobias.zinner@lmu.de)

**Abstract.** During the ACRIDICON-CHUVA aircraft campaign in September 2014 over the Amazon, among other topics aerosol effects on the development of cloud microphysical profiles during the burning season were studied. Hyperspectral remote sensing with the imaging spectrometer specMACS provided cloud microphysical information for sun-illuminated cloud sides. In order to derive profiles of phase or effective radius from cloud side observations vertical location information is indispensable. For this purpose, spectral measurements of cloud side reflected radiation in the oxygen-A absorption band collected by specMACS were used to determine absorption path length between cloud sides and the instrument aboard the aircraft. From these data horizontal distance and eventually vertical height were derived

It is shown that, depending on aircraft altitude and sensor viewing direction, an unambiguous relationship of absorption and distance exists and can be used to retrieve cloud geometrical parameters. A comparison to distance and height information from stereo image analysis (using data of an independent camera) demonstrates the efficiency of the approach. Uncertainty estimates due to method, instrument and environmental factors are provided. Main sources of uncertainty are unknown in-cloud absorption path contributions due to complex 3D geometry or unknown microphysical properties, variable surface albedo and aerosol distribution. A systematic difference of 3.8 km between stereo and spectral method is found which can be attributed to 3D geometry effects not considered in the methods simplified cloud model. If this offset is considered, typical differences found are 1.6 km for distance and 230 m for vertical position at a tpyical distance around 20 km between sensor and convective cloud elements of typically 1-10 km horizontal and vertical extent.

## 1 Introduction

Information on location and extent of clouds is central for any assessment of the role of clouds in the atmosphere. Knowledge of cloud vertical or lateral extent does not only allow for a first order estimate of the total water content, but also allows to estimate their contribution to the radiation balance of the climate system. In contrast to active remote sensing techniques providing an immanent distance measurement, passive remote sensing techniques need additional information sources in order to assign a location to the observed values. For satellite techniques, this is often achieved by use of observations in the thermal spectral range for the vertical. Based on the assumption that a cloud emits thermal radiation as a black body emitter, the observed

brightness temperature is interpreted as cloud top temperature which can be converted to a height if the temperature profile is known (e.g. Smith and Platt, 1978). More complex thermal techniques also relax the black body assumption to derive cloud heights for semi-transparent cloud layers from two or more thermal channel's observation differences (split window techniques, CO2 slicing, e.g.: Chahine, 1974; Menzel et al., 2008).

In addition to simple vertical cloud extent, other aspects of cloud geometry strongly affect passive remote sensing of cloud microphysical properties through illumination effects, e.g., shadows or bright cloud slopes (Varnai and Marshak, 2002; Zinner et al., 2006; Varnai and Marshak, 2007; Vant-Hull et al., 2007; Liang and Di Girolamo, 2013; Grosvenor and Wood, 2014). Especially on higher spatial resolution effects caused by geometry variation are of important influence (Zinner et al., 2006; Zinner and Mayer, 2006).

A method based on high resolution cloud reflectivity measurements in the solar spectral range is the so-called cloud side remote sensing approach, proposed by Martins et al. (2011) and Zinner et al. (2008) for the retrieval of cloud microphysical properties (particle size and phase) along the vertical profile of convective clouds. For this approach, Ewald et al. (2018) shows that cloud surface orientation can explain most of the observable variation of cloud reflectivity due to three-dimensional radiative transfer. If cloud surface orientation would be known, it could be considered for an important improvement of all

passive cloud remote sensing methods. For cloud side observations, this is also demonstrated in Ewald et al. (2015) using a scanning cloud radar to reconstruct cloud geometry.

For the cloud side retrieval of convective microphysics profiles, a localisation of each observation to a vertical position is obviously indispensable (e.g. for thermodynamic phase in Jäkel et al. (2017) or effective radius in Ewald et al. (2018)). Originally Martins et al. (2011) suggested to use a thermal cloud side imager to obtain the necessary information. Cloud radar

aided localisation as in Ewald et al. (2015) is also possible within its sensitivity limitations. Both methods depend on additional measurements and instrumentation with its own limitations and sensitivies. E.g., thermal imagery is affected by molecular absorption (especially water vapour), specifically when used at a slanted view through dense lower atmosphere. The necessary matching of multi-sensor observations introduces additional uncertainties.

During the German-Brazilian ACRIDICON-CHUVA (Aerosol, Cloud, Precipitation, and Radiation Interactions and Dy-

namics of Convective Cloud Systems–Cloud Processes of the Main Precipitation Systems in Brazil: A Contribution to Cloud Resolving Modeling and to the GPM (Global Precipitation Measurement)) aircraft campaign in Brazil 2014 (Wendisch et al., 2016) cloud side observations of solar reflectivity where collected with the imaging cloud spectrometer specMACS (Ewald et al., 2016). Spectral radiance between 420 and 2500 nm was measured through a side window of the German research aircraft HALO (High Altitude LOng range Gulfstream G550; Krautstrunk and Giez, 2012; see Fig. 1). Instead of using thermal

imagery or active measurements of distance, we will present the derivation of cloud distance based on the available spectral imager specMACS data itself. Oxygen-A absorption band measurements in the short-wave infrared around 760 nm provide distance information derived from the atmospheric absorption path. This way all data generated is provided on the same instrument specific coordinate system (time and space) and can be easily combined to provide products, e.g., the typical profile of particle size along the cloud side.

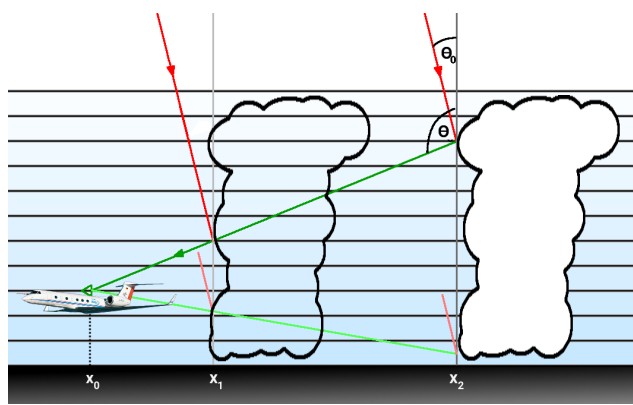

**Figure 1.** Illumination and observation geometry for cloud side remote sensing. A sensor mounted on an aircraft at height $z$ observes a cloud side at distance $x_i - x_0$. Solar radiation enters the atmosphere at a solar zenith angle $\theta_0$ to the cloud where it gets scattered and reaches the sensor at sensor zenith angle $\theta$. Dependent on cloud distance, radiation within the oxygen absorption band gets absorbed on this path according to the path length and increasing atmospheric density with decreasing height. This 2D illustration shows a situation with the solar illumination in line with the sensor viewing direction (relative sensor azimuth $0°$). In this study relative sensor azimuth $\pm 45°$, solar zenith $0 < \theta_0 < 30°$, and sensor zenith angles $40°$ (below horizon) $< \theta_0 < 100°$ (above horizon) will be considered.

The derivation of distance from the absorption along the light path of reflected light in the oxygen A-band was originally proposed by Yamamoto and Wark (1961) for satellite application. Wu (1985), Fischer and Grassl (1991) and Fischer et al. (1991) discussed the theory behind the approach in detail. Recently Yang et al. (2013) nicely summarize different approaches to the retrieval of cloud height and cloud thickness at the same time by combination of measurements of the DISCOVR EPIC sensors in the oxygen A- and B-bands. Merlin et al. (2016) suggest to derive both parameters from a combination of oxygen-A measurements at different angles. Depending on cloud type and the choice of spectral information used, an accuracy between 50 and a few hundred metres is found. We adapt the method to cloud sides and near horizontal light paths between sensor and clouds for the specMACS sensor. In contrast to other approaches, as e.g. a stereo image analysis, this method is not limited to a few features of high contrast, but provides distance over large continuous parts of the available data.

Nevertheless data from a manual stereo distance analysis by Jäkel et al. (2017) for a number of ACRIDICON-CHUVA cloud cases is used for comparison to the absorption path derived distance. Stereo techniques were first applied to clouds from operational geostationary and polar-orbiting satellites (Hasler, 1981; Muller et al., 2007). For high spatial resolution ground-based stereo-camera systems Seiz et al. (2007) and Beekmans et al. (2016) estimate typical distance biases in the order of a few 100 m or a few percent of the distance. Opposed to the usual use of two pictures synchronously taken from two fixed cameras, Jäkel et al. (2017) used consecutive pictures from one and the same camera onboard the moving HALO aircraft.

Distance determination for the ACRIDICON-CHUVA cloud side observations is presented in the following. A detailed analysis of uncertainty is conducted for the oxygen-A method applied to our sensor setup and an uncertainty estimate product is provided as part of the method. Geometrical heights obtained this way are compared to stereo analysis results.

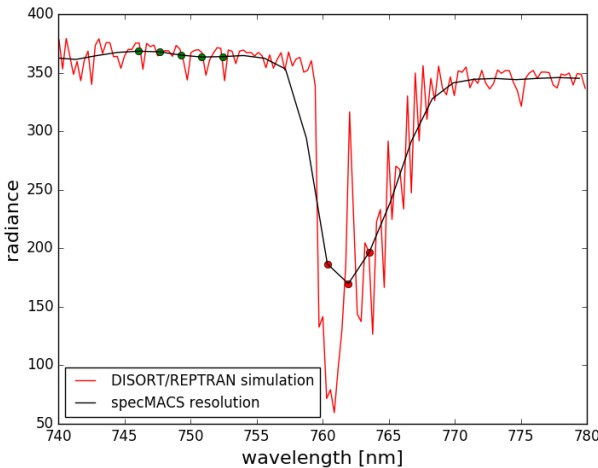

**Figure 2.** DISORT simulation of oxygen-A band using on REPTRAN medium resolution (0.2 nm, red line) and reduced to specMACS spectral response (black line, sampling 1.6 nm, FWHM = 2.8 nm). Colored dots label specMACS channels used in this work.

## 2 Distance retrieval from oxygen A-band absorption path

### 2.1 Measurements and modelling of spectral radiance

Through a side window of the German HALO (High Altitude LOng range) aircraft reflected sunlight around the oxygen A-band was measured by the imaging cloud spectrometer specMACS during the ACRIDICON-CHUVA campaign (Fig. 1). It has
a spectral resolution of 2.5-4 nm below 1000 nm and 7.5-12 nm above 1000 nm wavelength. Data was collected at a spectral sampling of 1.6 nm and a spectral bandwidth of 2.8 nm (FWHM) in the region of the oxygen A-band (Ewald et al., 2016). Ewald et al. (2016) characterised spectral channel positions as well as the spectral filter function width in detail.

Figure 2 shows the spectral shape of the oxygen A-band in reflectivity at specMACS sampling together with a simulation at higher resolution. These simulated spectra, as well as all radiative transfer calculations necessary for the setup of the method
were done with the libRadtran package (Mayer and Kylling, 2005; Emde et al., 2016). The measurement setup shown in Fig. 1 with a sideward viewing sensor, near horizontal absorption paths, and nearly vertical convective cloud sides can not be represented in a one-dimensional plane-parallel standard setup. Thus, the 3D radiative transfer model MYSTIC (Monte Carlo code for the physically correct tracing of photons in the atmosphere Mayer et al., 2009) was applied. Spectral gas absorption was considered using the REPresentative wavelength radiative TRANsfer method (REPTRAN) by Gasteiger et al.
(2014). REPTRAN is based on line-by-line calculations. Radiances are approximated as a weighted mean at representative wavelengths. As mentioned above, Figure 2 shows the spectral region of the oxygen-A band at different spectral resolutions. The REPTRAN medium resolution (about 0.2 nm in this spectral range) was found to provide sufficient accuracy compared to higher spectral resolution and will be used in the following. At this resolution still part of the detail of the absorption band

is visible. Details like the two absorption minima at 761 and 763 nm are not resolved at specMACS resolution (black line in Fig. 2). Still a number of channels can be used to provide a valid measurement of absorption band depth. Five specMACS channels between 745 and 754 nm (green dots) will be used as reference measurement not affected by oxygen absorption and three channels inside the absorption band between 759 and 765 as a measure of absorption (red dots).

## 2.2 Retrieval

After the solar radiation is reflected by clouds the absorption path through the atmosphere mainly depends on the distance and the observation zenith angle $\theta$. While $\theta_0$, $\theta$ and the general oxygen distribution in the atmosphere are known, cloud distance can be derived from the absorption signal within some limitations. Fischer and Grassl (1991) propose a cloud height derivation based on a spectral measurement around the oxygen A-band with a resolution on the order of 1 nm, but also demonstrate that lower spectral resolution of several nanometers could be sufficient. Technically, they use the relation of cloud distance and a radiance ratio composed from one measurement next to the oxygen absorption region and one within the absorption band. In the following we describe our implementation of this approach and the expected sources of error and uncertainty.

### 2.2.1 Sensitivities

The impact of a range of sources of uncertainty is investigated using radiative transfer simulations in the following. Already Fischer and Grassl (1991) emphasise the importance of a knowledge of the spatial distribution of scattering coefficient or cloud optical thickness. Both influence the in-cloud photon absorption path and thus the cloud reflectance within the absorption band. We plan to derive distance only for optically thick clouds and thus expect to be in a saturation regime for reflected radiances which would minimise errors due to unknown optical thickness. The unknown 3D distribution of the scattering coefficient (related to cloud microphysics) still could strongly affect the method - influence of liquid water content (LWC), cloud particle size and cloud extent are tested. Neglected detail of the aerosol distribution could have similar effects. Surface albedo and pressure profile variations are tested as well as the impact of the spectral calibration accuracy of the used spectrometer.

Figure 1 shows the setup for the 3D Monte Carlo simulations in this section. The horizontal cloud extent in the following is $\Delta x = 2km$, cloud liquid water content is at a homogeneous 0.5 $\mathrm{g/m^3}$, and effective radius at constant 10 $\mu m$. Example oxygen-A spectra are shown in Fig. 3. The dependence of the absorption band depth on distance (Fig. 3a) will be utilised to retrieve cloud distance - and together with observation zenith angle $\theta$ - will provide cloud vertical position. In the following "distance" is always the horizontal distance between sensor and cloud (as opposed to the line-of-sight distance).

All spectra in Figure 3 are normalised to the minimum value at the 762 nm channel in order to visualise the band depth. As introduced above, the absorption signal is reduced to a radiance ratio similar to Wu (1985) and Fischer and Grassl (1991). Following their considerations and the fact that specMACS lacks spectral resolution compared to more specialised sensors, we average all three channels available within the oxygen band (red dots) and five channels next to it (green dots). We define the oxygen-A absorption ratio

$$R_{\mathrm{O2A}} = I_{759-764\mathrm{nm}}/I_{745-754\mathrm{nm}}. \tag{1}$$

The larger the absorption, the smaller the ratio is. Fig. 3 gives the values of $R_{\mathrm{O2A}}$ related to the shown absorption band depths.

Although spectral position and spectral filter functions have been characterised by Ewald et al. (2016), some remaining uncertainty has to be assigned to this calibration: An accuracy of the spectral position of about 0.1 nm and an uncertainty of the spectral width of 0.2 nm can be assumed (Ewald et al., 2016). As a test of the impact, a variation of bandwidth of $\pm\,0.2$ nm is applied. It yields small differences of around 1% in the observed radiance ratio as shown in Figure 3b.

Absorption path length depends on sensor zenith angle $\theta$, as it controls the total absorber amount to be passed after scattering by the cloud in combination with the observer height (Fig. 3c). Changes of solar zenith angle and relative azimuth angles have a much smaller effect on the path length (Fig. 3d,e). Using aircraft orientation data these parameters can be characterised with sufficient accuracy.

Other characteristics of atmosphere and surface situation are not as accessible. Higher surface pressure would lead to an increase of oxygen absorption and a decrease in $R_{\mathrm{O2A}}$ at a given height. It is obvious in Fig. 3f that this influence is small for the observed range of surface pressure during the ACRIDICON-CHUVA campaign (995 - 1015 hPa). Surface albedo can also influence the absorption band depth due to multiple scattering between surface and atmosphere. Longer absorption pathes are generated for larger surface albedo (Fig. 3g). Aerosol concentration along the path influences the path length in a similar way: The higher the aerosol content, the longer the path due to multiple scattering, and the smaller $R_{\mathrm{O2A}}$ (Fig. 3h).

The spatial distribution of scatterers in clouds influences the in-cloud path lengths. Wu (1985) or Fischer and Grassl (1991) point out that the spatial distribution of the scattering coefficient as well as the optical thickness of the cloud layer are critical for their retrievals of cloud height for a wide range of optical thick and thin clouds. In case of cloud side remote sensing of convective cloud microphysics the task is simplified. The object of interest is a dense vertical cloud with a horizontal spatial extent of the order of kilometres. A liquid water content of 0.5 $\mathrm{g/m^3}$ (typical for ACRIDICON-CHUVA Wendisch et al., 2016) and an extent of just 1000 m already leads to large (horizontal) optical thickness values in the order of 50 and above. Wu (1985) shows that these circumstances minimise uncertainties. Figures 3i, j and k show the effect of varying characteristics of the cloud on the absorption band depth: LWC, particle size and horizontal extent are varied. Cloud extent does not cause any important uncertainty, influence of droplet size is also small. Unknown LWC leads to a larger potential uncertainty especially for low water content.

Figure 3m demonstrates the effect of a deviation of the general cloud geometry from the idealised "cloud wall". Absorption band results are simulated for a large spherical cloud here. A cloud sphere with diameter of 2 km is placed at fixed 10 km horizontal distance. In order to vary the angle of the cloud surface relative to the horizon without changes of sensor elevation angle or distance, its vertical and horizontal position is changed (vertically and with respect to distance to sphere center). $\theta_{\mathrm{cloud}} = 90°$ is comparable to the reflected radiance from a vertical cloud surface as shown in the other displays before. The surface tilts away from the vertical towards the cloud top until it reaches a horizontal surface with $\theta_{\mathrm{cloud}} = 0°$. The more horizontal the cloud surface is, the more likely short photon pathes between sun, cloud and sensor become. A retrieval based on the assumption of vertical cloud sides would provide a negative distance bias. An important consequence of this last test is the limitation of the method developed in the next section to cloud sides. Cloud top areas as well as horizontal cloud formations,

like stratocumulus decks, should be excluded. It can be expected that an average deviation from the assumption of vertical cloud sides will lead to a systematic effect.

### 2.2.2 Lookup-table

The retrieval of sensor-cloud distance will be realised through forward simulations describing the relation of absorption ratio $R_{\mathrm{O2A}}$ and distance. As mentioned before, the geometry of the observation situation requires 3D radiative transfer simulations. As these are very time consuming, strong constraints apply to the affordable computational effort. This leads to the simplification of convective cloud geometry to vertical "cloud walls". Consideration of different or more typical cloud side inclination would be possible in principal – e.g., a cloud surface slightly tilting away from the vertical. However, already at this stage, the derivation method is based on extensive time consuming 3D radiative transfer simulation. Unfortunately the implementation of tilted cloud slopes into an orthogonal x-y-z simulation grid would increase the computational effort to the point of futility, because calculations on a much finer spatial grid would become necessary.

The horizontal distance and the observation geometry (sensor height, sensor zenith angle, solar zenith angle) define the dimensions of a lookup table. All other sensitivities will be neglected and have to be considered as source of uncertainty to be quantified: either because parameters can not be constrained at all (LWC, $r_{\mathrm{eff}}$, cloud horizontal extent, local aerosol and albedo situation), or uncertainties introduced are so much smaller than the above mentioned that the expensive extension of the lookup table with another free parameter seems unnecessary (surface pressure, solar azimuth angle). A detailed uncertainty analysis is provided later.

To generate a lookup table forward simulations are set up for the selected environmental parameters listed in Table 1. Observation zenith angle $\theta$ are discretized in steps of 2° between 41° (below horizon) and 99° (above horizon), cloud distances at 17 values of increasing distance between 0.5 and 60 km, and flight altitudes in steps of 1 km between 2 and 11 km. The cloud for these simulations has the dimensions $\Delta x = 2 km$ (in viewing azimuth direction), $\Delta y = \infty$ (perpendicular to viewing azimuth angle) and $\Delta z = 12 km$ (between surface and 12 km height).

For combinations of these parameters reflected radiances are simulated using libRadtran/MYSTIC with REPTRAN medium. Eight simulated specMACS channels (as labelled in Fig. 3) contribute to the ratio $R_{O2A}$ as defined in Eq. 1.

Figure 4 shows a sub-set of these forward simulations ($\theta_0 = 30°$, $z_s = 6$ km). For increasing distance to the cloud and for several sensor zenith angles $\theta$ the ratio $R_{\mathrm{O2A}}$ is displayed. For a given sensor height the cloud wall is visible under certain observation zenith angles only up to a certain distance. E.g., for $\theta = 73°$ (17° below horizon, blue color) the cloud is visible only up to about 16 km. Only the surface is visible at the same viewing direction for greater distances. For a sensor zenith angle above horizon ($\theta > 90°$) only the sky is visible for large distances.

The 3D radiative transfer simulations used to create the results are subject to Monte Carlo noise, observable as jiggle on the black line connecting Monte Carlo results. The model provides exact results within the uncertainty allowed by the photon statistics of the model. Here the standard deviation of reflectivity results is 1.5%. A reduction of this intrinsic uncertainty would be possible through an increase of the simulated number of photons $N$. As uncertainty decreases according to $\sqrt{N}$, the required increase in computational time would be prohibitively large. Already for this accuracy, simulation of 20,000 combinations (4

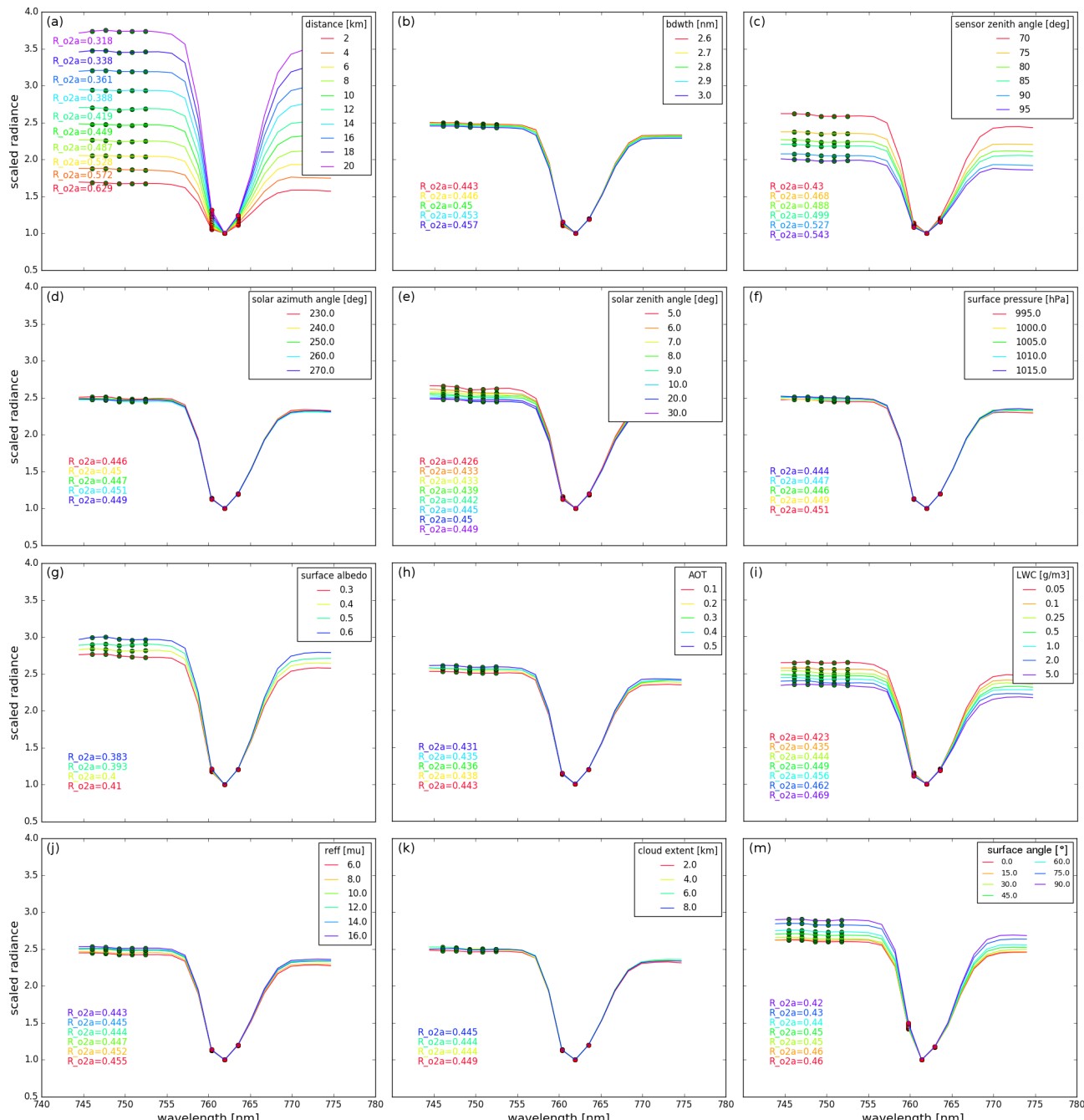

**Figure 3.** Sensitivity of the oxygen-A absorption band at specMACS resolution to different parameters. MYSTIC Monte Carlo radiative transfer simulations for a "cloud wall" setup (for a-k) or a "cloud sphere" (m). The basic situation is characterised by solar zenith angle $\theta_0 = 30°$, observation zenith angle $\theta = 75°$ ($15°$ elevation below horizon), relative sensor azimuth angle between solar and observation azimuth $0°$, sensor altitude $z_s = 6km$, distance to cloud $\Delta x = 10km$, horizontal cloud extent $\Delta x = 2km$, cloud liquid water content 0.5 g/m$^3$, effective radius 10 $\mu m$, no aerosol, albedo $a = 0$, tropical standard atmosphere with 1000 hPa surface pressure (Anderson et al., 1986). Starting from this setup single parameters are varied: distance to cloud (a), specMACS spectral bandwidth (b), observation zenith angle (c), relative solar azimuth angle (d), solar zenith angle (e), surface pressure (f), surface albedo (g), aerosol optical thickness (h), cloud liquid water content (i), cloud effective radius (j), horizontal extent (k) and cloud surface angle relative to observation (m). The five specMACS spectral channels next to the absorption band labelled with green points and the three within the absorption line are used in the retrieval.

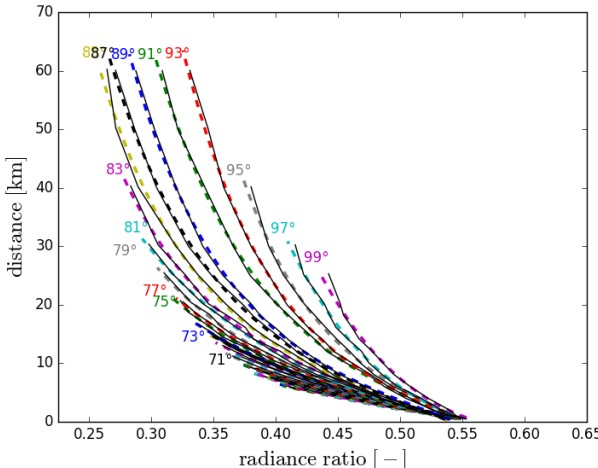

**Figure 4.** MYSTIC Monte Carlo simulations for a "cloud wall" geometry. Example for the lookup-table forward simulations showing the dependence of cloud distance and reflected radiance ratio $R_{O2A}$ for $\theta_0 = 30°$, $z_s = 6$ km (thin black lines). Varying sensor zenith angle $\theta$ is shown in colour (horizon at $90°$). For each $\theta$ value and each solar zenith angle $\theta_0$ a polynomial fit to Monte Carlo results $p_{\theta,\theta_0}$ is obtained and used to derive distance in the retrieval: $d = p_{\theta,\theta_0}(R_{O2A})$ (broken coloured lines). Shown are results for all $\theta$ between $41°$ (below horizon) and $99°$ (above horizon) for which the cloud wall between surface and 12 km height is visible at the respective distance.

different solar zenith angles, 10 different sensor heights, 17 distances, and 30 sensor zenith angles) consumes more than one week of computation time on a large cluster machine. Substantial improvement, e.g. reduction to 0.15%, would require an increase of this time by a factor of 100. Instead, we limit the impact of the noise by using a third order polynomial fit on the Monte Carlo simulations.

Figure 4 shows these individual fits as dashed coloured lines. Furthermore this reduction of the individual results to a set of polynomial functions $p_{\theta,\theta_0}$, one for each sun and sensor geometry, allows for an effective retrieval using these functions to derive distance $d = p_{\theta,\theta_0}(R_{O2A})$. Although the impact of Monte Carlo uncertainty is much reduced through the fitting, it still results in a contribution to the uncertainty to be considered.

With given values of solar zenith angle $\theta_0$, sensor zenith angle $\theta$ and flight altitude this set of polynomial functions provides

a distance retrieval for each $R_{O2A}$ measurement. Distance can be retrieved for situations with solar zenith angle between 3 and about $60°$. The allowed relative azimuth between sun and sensor view is limited to $45°$. Distance derivation for larger azimuth differences is strongly affected by 3D effects (e.g. shadows). By filtering out cloud tops, which are always more or less horizontal, the impact of non-vertical cloud surfaces will be minimised.

### 2.2.3    Uncertainty budget

Table 1 summarises observed ranges for several environmental parameters during ACRIDICON-CHUVA. As these are either not known or not represented in the forward simulations, they lead to uncertainty for the retrieval based on the oxygen-A ratio

**Table 1.** Summary of uncertainties due to environmental parameters during ACRIDICON-CHUVA not fully resolved through forward simulations. Each parameter is given, its observed range of values (wherever available), the values considered in the lookup table of forward simulations, and the estimate of the remaining uncertainty based on sensitivity tests as shown in Fig. 3. Given is a an estimation of related relative standard deviation $\sigma_R$ derived from the difference provided by the observed values listed.

| parameter | observed values | values LUT | $\sigma_R$ [%] |
|---|---|---|---|
| surface pressure | [1] 997 - 1005 hPa | 1000 hPa | 0.3 |
| aerosol OT | [2] 0.07 - 0.55 (mean 0.22) | 0.2 | 0.8 |
| albedo at 760 nm | [3] 0.25 - 0.5 (mean 0.35) | 0.35 | 1.3 |
| solar zenith angle | 3-60° | 5,7,10,30° | 1.0 |
| relativ azimuth angle | 180 ±45° | 0° | 0.3 |
| LWC | [4] 0.1-2.0 g/m$^3$ | 0.5 g/m$^3$ | 1.8 |
| $r_{\text{eff}}$ | [4] 6-20 µm | 10 µm | 0.9 |
| cloud extent | [5] 1-10 km | 2 km | 0.3 |

Data from [1] radiosonde data for Manaus airport, [2] AERONET AOD at 675 nm for stations: Manaus EMBRAPA, Rio Branca, Ji Parana, Alta Floresta (mean over all stations), [3] MODIS 16-day albedo product, [4] taken from in-situ measurements during campaign, [5] typical values.

$R_{\text{O2A}}(= R$ for simplicity in the following). Using these parameter ranges together with sensitivity tests as shown in Fig.3 uncertainties for the ratio are derived. A maximum spread for $R_{\text{O2A}}$ can be derived from the observed values, e.g. for pressure $R_{\text{min}} - R_{\text{max}} = R(1005 \, \text{hPa}) - R(997 \, \text{hPa})$. In specific observation situations even more extreme local values could be present. Assuming that $R_{\text{min}}$ and $R_{\text{max}}$ at least limit the range including 95% of all possible values in a Gaussian error distribution, we

can simplify our error estimation. A standard deviation value might then be defined as

$$\sigma_{\text{R}} = \frac{1}{4} \left( R_{\text{max}} - R_{\text{min}} \right) / R_{\text{LUT}}, \tag{2}$$

describing the uncertainty in a specific observation situation due to environmental parameters. If it is further assumed that all uncertainties in Table 1 are independent, we can use error propagation and take the square root of the total of all squared contributions to find an overall uncertainty estimate for the ratio $R_{\text{O2A}}$ of about $\sigma_R = 2.8\%$.

These environmental uncertainties are combined with the mentioned Monte Carlo uncertainties and sensor calibration accuracy (spectral and radiometric) to provide a total error budget. For three different observation altitudes Figure 5 shows an example of the resulting uncertainty budget for derived distance from all sources apart from 3D effects (which are limited by suited filtering of the observation scenes). The shape of these maps with respect to distance and elevation angle is caused by the general cloud/observer geometry. At low altitudes most cloud sides can be observed looking upward, while for high altitudes

they are observed looking downward. Relative uncertainties of distance become large (>15%) for distances smaller than 5 km and for viewing zenith angles close to or above the horizon at 90°. The reasons are the unknown cloud parameters and albedo: If the sought distance signal on absorption is weak, either due to small cloud distance, or due to a shortage of absorbing oxygen (e.g. more horizontal compared to more downward looking observation geometries), the impact of these unknown boundary

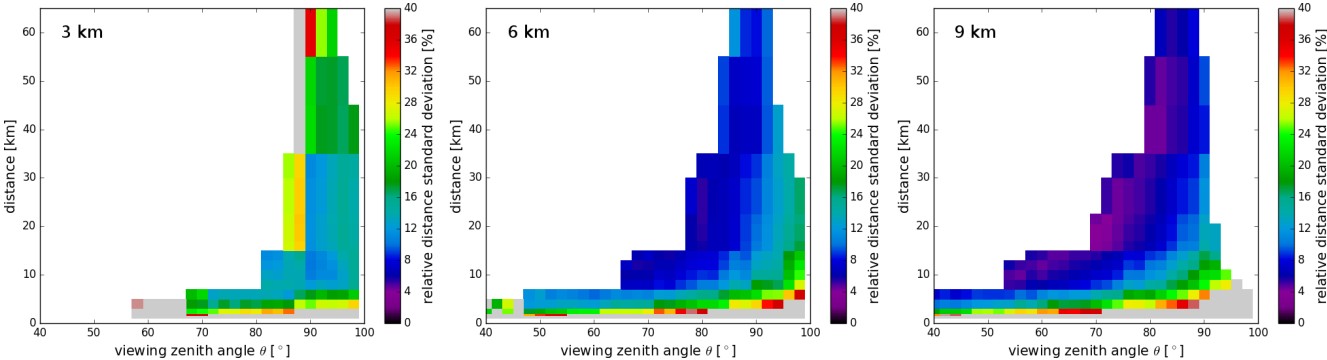

**Figure 5.** Calculated standard deviation of oxygen-A derived distances due to uncertainty of Monte Carlo simulation, sensor calibration, environmental parameters and their discretization in the lookup-table of forward simulations (compare Tab. 1) depending on cloud distance and observation zenith angle for three of the tabulated observation altitudes.

conditions is stronger. Effects of aerosol situation, Monte Carlo statistics, and sensor accuracy are much smaller. Only at low altitudes the higher aerosol content in these layers has larger impact.

Uncertainties of more than 10% might be too large for most applications of distance measurement.This is also true for the application to cloud side remote sensing intended here (Jäkel et al., 2017; Ewald et al., 2018). The geometrical setup of our application, the vertical localisation of cloud microphysical observations from a flying aircraft, fortunately improves the situation: Relative uncertainty of 10% in horizontal distance maps onto the same relative uncertainty in altitude difference using the observation elevation angle. However, the distance affected this way is only the altitude difference between the point on the cloud and the observer, i.e., the flight altitude. For a cloud altitude of 7500 m and a flight altitude of 5000 m, a relative uncertainty of 10% in distance translates into an altitude uncertainty of only 250 m, or 3%. In the next step height retrieval and connected uncertainty budget will be systematically compared to independent measurements of height.

## 3 Comparison to stereo derived distance and height

In order to check the validity of distance and height measurements derived from the oxygen-A band, we compare this data from the specMACS imager to completely independent stereo cloud distance estimates from a 2-D camera system. specMACS observes at a pushbroom geometry to the side, while stereo-matching depends on the tracking of the same image features for different angular position on 2D pictures. The camera used for the latter is a GoPro Hero (manufactured by GoPro, Inc., USA, HD3+3660-023 Full-HD, hereafter GoPro) which was fixed to a side window next to the specMACS sensor during ACRIDICON-CHUVA. It collected RGB images of 1920x1080 pixels size with a total field-of-view of about 90°x60° at a frame rate of 30 Hz. The cloud sides observed by specMACS were visible in the 2D imagery due to GoPro's much wider field-of-view along and across-track. In order to compare results for cloud distance from both systems, an accurate temporal and geometrical matching of the data sets is necessary. Automation of this matching is tedious, in particular because mounting

and time registration of the GoPro camera changed between flights. Therefore Jäkel et al. (2017) did a manual stereo-analysis for several cloud cases during the ACRIDICON-CHUVA flights. They matched time and space coordinates of both systems for selected traceable cloud side features and derived distances and heights for their remote measurements of cloud phase. In the following, results of their stereo derivation for 500 stereo points over 27 cloud cases and the automatic oxygen-A absorption based cloud height maps are compared.

For high spatial resolution ground-based stereo-camera systems and distance between observer and cloud of 4 - 10 km, Seiz et al. (2007), Öktem et al. (2014) and Beekmans et al. (2016) estimate typical distance biases in the order of a few 100 m or a few percent of the distance with larger errors for larger distances. Jäkel et al. (2017) estimate their accuracy to be around 200-300 m. Reason for these accuracy limitation are, e.g., incomplete or variable camera orientation information (e.g. for the Jäkel et al. (2017) analysis), uncertainties in camera distortion characterization, camera angular resolution limitations, limited aircraft orientation accuracy or wind drift.

With these values and our analysis shown in Figure 5 it seems likely that oxygen-A retrievals produce similar values for clouds up to 10 km distance. For more distant clouds (and especially for distances above horizon and for strong 3-D effects) the uncertainty of oxygen-A retrieval uncertainties are larger than these values. However, for distances above 10 km stereo methods uncertainties are most likely increasing too, as camera projection uncertainties in these methods lead to distance dependent uncertainty contributions. Figure 6 shows a comparison of oxygen-A derived distances and stereo derivations from Jäkel et al. (2017) for an example case. A systematic comparison to several hundred stereo derived positions is shown in Figure 7.

One minute of specMACS data at 745.5 nm wavelength from ACRIDICON-CHUVA flight AC18 on 28-09-2014 around 18:51 UTC at an altitude of 9.1 km is shown in Figure 6a. Data was collected by specMACS along the time axis with a $32°$ spatial field-of-view across track (dark current calibration at around 18:51:10 UTC). The sensor is looking sideward centred at $5°$ below horizon with respect to the aircraft orientation (here $85°$ sensor zenith angle). The solar zenith angle was $37.5°$ and the relative azimuth angle $177°$. That means, the sun was almost exactly in the back of the observing sensor. Figure 6b shows the oxygen-A ratio $R_{O2A}$ for this case. Obviously the measured ratio is strongly depending on distance. While nearby clouds at low sensor zenith angle show large values, corresponding to short absorption path lengths, distant clouds towards horizon show much smaller $R_{O2A}$ values due to much larger absorption path lengths. Cloud tops have the tendency to display shorter absorption paths and larger ratios due to 3D effects. In a similar way flat horizontal surfaces display shorter absorption pathes than vertical cloud parts. Likely reason is the 3D cloud surface orientation effect as deomstrated in Fig. 3m. Shadows are rare at this solar illumination and they are only casted on the lower cloud parts (e.g. central cloud element at $75°$ sensor zenith angle). If present, they display relatively long absorption paths and smaller ratios than the rest of the cloud, because these parts are only illuminated by indirect, scattered light. Using the method described in Section 2.2 values of $R_{O2A}$ can now be attributed to distances depending on observation zenith angle . Then distances are translated into vertical height values, using the flight altitude (here 9.1 km) and observation zenith angle (Fig. 6d).

Distances from the Jäkel et al. (2017) paper are also shown in Fig. 6. 19 stereo points derived from GoPro imagery could be identified (by eye) in the data. The distance and height from Jäkel et al. (2017) are shown in kilometres together with an

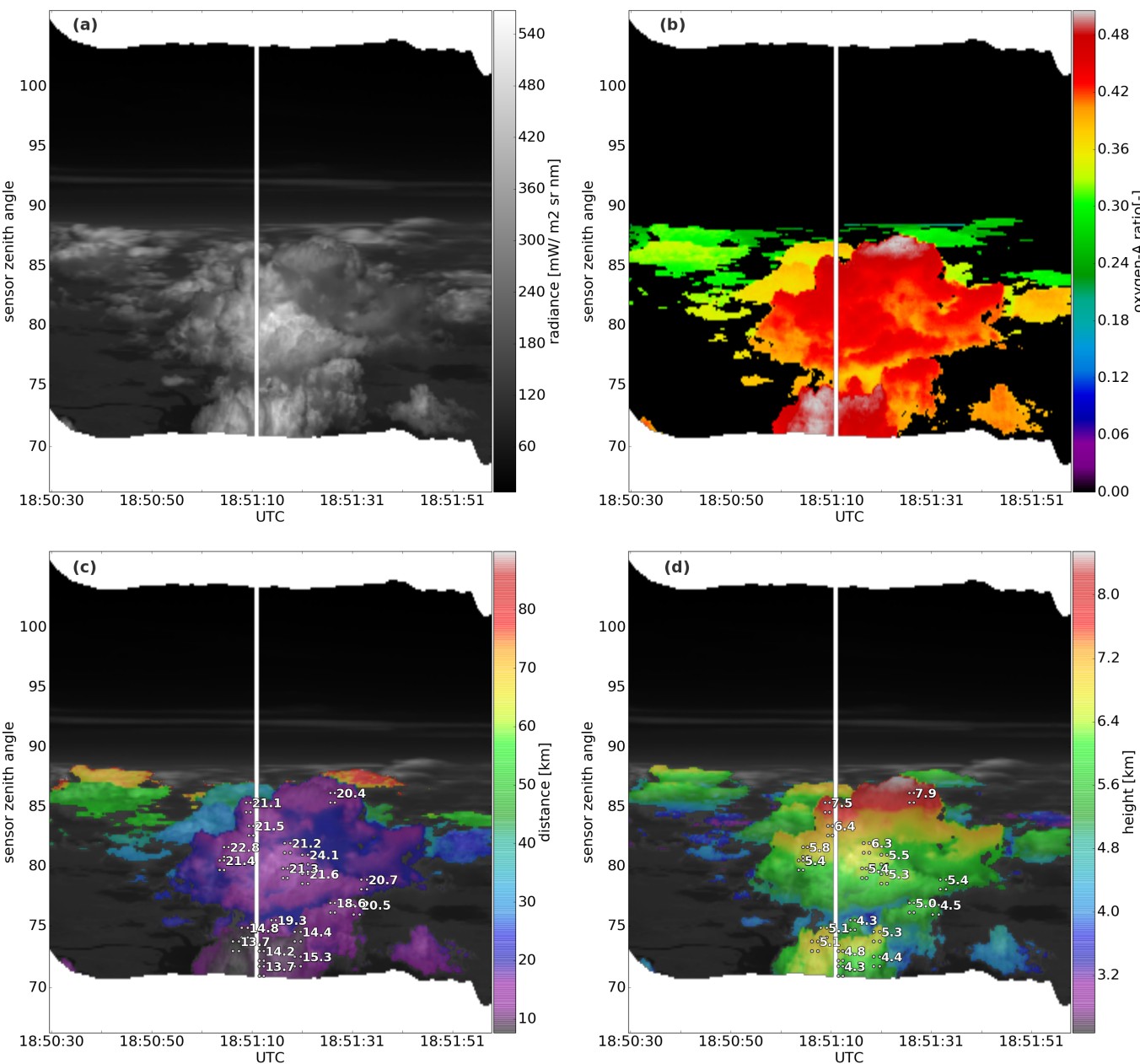

**Figure 6.** specMACS data from ACRIDICON-CHUVA flight AC18 on 28-09-2014 (flight altitude 9.1 km). (a) visible radiance image at 750 nm wavelength, (b) oxygen-A absorption ratio $R_{\text{O2A}} = I_{759-764\text{nm}}/I_{745-754\text{nm}}$, (c) distance derived from $R_{\text{O2A}}$, (d) height derived from $R_{\text{O2A}}$. In (c) and (d) values from stereo analysis of distance and height from Jäkel et al. (2017) are included; points label regions of oxygen-A derived values compared to the given stereo derived values in Fig. 7.

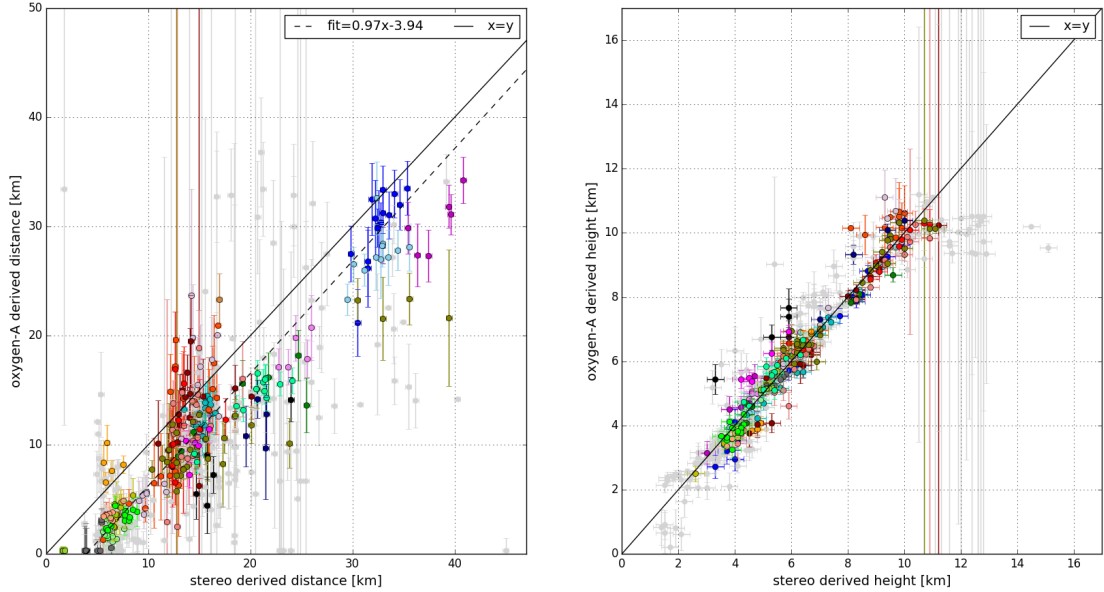

**Figure 7.** Systematic comparison of specMACS oxygen-A derived cloud distance and height to GoPro stereo derived distance and height from Jäkel et al. (2017) for 519 points and 27 cloud situations (27 different colors). Turquoise points represent the valus from Fig. 6. (a) shows distance values (233 points filtered out, 286 used in quantitative analysis), error bars for stereo values are ±300 m, error bars on oxygen-A derived distances show the error budget provided by retrieval (cf. Fig. 5 and Sec. 2.2.3) combined with the variation within oxygen-A sections (cf. Fig.6). (b) shows height values for both method after a 3.7 km distance offset correction was applied to all points. Black lines show 1:1 line, broken black line in (a) is result of a linear polynomial fit through 286 points.

area of oxygen-A data to which they are compared to (represented by 4 corner points). Values are compared for these areas instead of exact positions, as the accuracy of a matching of points in one observation geometry (GoPro, 2D angular imagery) to a very different geometry (specMACS, one angular and one time dimension) is limited. Area size is selected to represent about 200 m × 200 m at 10 km distance to the cloud side. The 19 stereo values in Figure 6c range from 13.7 to 24.1 km. The related oxygen-A distances are somewhat lower between 10 and below 20 km distance. For the reasons mentioned earlier, the difference between the derived cloud heights from stereo and oxygen-A data in Figure 6d is less striking.

The systematic analysis in Figure 7a reveals that the tendency to underestimate stereo distances is typical for many cases. From 27 cloud scenes 519 stereo points were identified within specMACS data. 233 of these are not considered for quantitative comparison (grey points, removed manually) , because they are located very close to cloud tops (e.g. the top two points in Fig. 6), they are located at angles more than 15° below horizon (four lowest points in Fig. 6), cloud geometry observed was a horizontal cloud deck and not a cloud side, or they are located in cloud shadows. The 286 coloured points in Figure 7a remain

for analysis. Data points from the above example show up in a bright turquoise colour close to the centre of the figure. 27 different colors stand for the 27 cloud scenes analysed.

The group of points filtered out is not only responsible for most of the larger deviations, but also shows the large error estimates provided directly by the oxygen-A retrieval. Apparently we filtered the correct points, difficult to retrieve for different reasons, before quantitative analysis. The remaining points line up along the 1:1 line with a noticeable offset for a majority of the points. The oxygen-A method seems to underestimate distance. If a linear polynomial fit is applied to all remaining points, an almost perfectly parallel line with a slope parameter of 0.97 and an offset of 3.76 km is found. These parameters do not strongly depend on the point filtering applied.

After careful analysis of other possible sources, we conclude that this offset of about 3.8 km towards shorter distances in the oxygen-A method is mainly caused by the strength of the typical 3D effects not considered in the retrieval. Order of magnitude and sign of the effect is consistent with the sensitivity tests using a cloud sphere in Figure 3m. A constant offset would be a consequence of systematically shortened absorption pathes due to the dominating geometry in the observed cloud scene. As a consequence, for the cloud height derivation in these ACRIDICON-CHUVA flights a simple distance offset correction of 3.8 km to all distances derived from oxygen-A data seems reasonable. This approach will not produce bias-free results for each individual scene as cloud geometry and viewing perspective vary between scenes. Figure 7a shows groups of coloured points (different cloud scenes) with different offsets. Still for these a partial bias will remain. Mostly in the order of 1-2 km. For other cloud types the approach would have to be adapted (e.g. more horizontal cloud layers) or different observer perspective (e.g. ground-based). Without auxilliary means of distance determination, this "calibration" step could be achieved from a small set of 3D forward radiative transfer simulation results for synthetic cloud cases, e.g., from cloud resolving modelling. This possibility and our conclusions regarding the role of 3D effects are illustrated in the discussion and conclusion section.

If this offset is applied, 60% of all 286 oxygen-A derived values coincide with stereo values within the shown error bars. A perfect consideration of all error sources in these error bars should have resulted in a share of 68% of all values agreeing within $\pm$ one standard deviation. This substantiates a basically correct consideration of all important error sources influencing accuracy of the retrievals and of the mapping of GoPro positions on specMACS data.

As last step of this method, distance has to be translated into cloud height using the flight altitude and the sensor zenith angle. As pointed out before, differences are diminished during this step due to the fact that only the vertical difference between cloud and flight altitude is affected by the uncertainties. This becomes obvious in Figure 7b. After offset correction, the comparisons points closely line up along the 1:1 line. 86% of all height value comparisons coincide within the uncertainty expectations, which implies that real uncertainties for the cloud height might even be smaller than the error budget predicted. The remaining standard deviation assuming a Gaussian distribution for the difference between stereo and oxygen-A derived distances is 490 m. The number is strongly influenced by a few large "non-Gaussian" outliers. The median absolute difference value, the typical difference of a single point, is 11% (1.6 km) for distance and 4% (230 m) for height.

## 4 Conclusions and discussion

The instrumentation of the ACRIDICON-CHUVA HALO campaign lacked a method to provide cloud localisation important for the application of cloud remote sensing data. Thus, a method was presented to derive cloud distance and vertical position of cloud elements from sideward viewing observations in the oxygen-A band using data from the specMACS imaging spectrometer. A distance derivation from cloud side reflected radiance was presented. Using this method, a straight forward possibility is provided to assign a position in space to all products derived from the same sensor. First main goal is the provision of a height value to generate cloud particle size or phase profiles averaged over whole convective clouds or cloud ensembles, a second goal the provision of a cloud surface orientation to be used to reduce the impact of 3D radiative effects on microphysics retrievals (see Jäkel et al., 2017; Ewald et al., 2018).

Uncertainties of the method have been characterised by radiative transfer experiments. The validity of the derived distance and height values and of their error budget calculations have been corroborated by comparison to a substantial number of stereo derived values from Jäkel et al. (2017). We find an average offset between both data sets of 3.8 km. If this offset is corrected, typical observed differences lie in the range expected from the error budget. Observed differences from the stereo derived values are somewhat larger which could be due to uncertainties in the stereo deviation itself.

For the large offset we simply use as a correction factor we consider 3D cloud geometry itself to be the most likely reason: cloud side geometry is not considered completely correct in the retrieval method due to computational limitations. Starting point of the derivation method is the simplification of cloud geometry into vertical cloud walls. A cloud side tilted away from the vertical towards the horizontal decreases the typical in-cloud oxygen absorption path contributions compared to a vertical cloud. This is especially noticeable approaching (horizontal) cloud tops, but is true to some extent for most real, not perfectly vertical cloud sides. On average over all cloud situations and distances, the value of the reduction of absorption path for the observed convective cloud fields with respect to the simulated absorption path is surprisingly constant around 3.8 km. Other candidates for the observed offset are albedo or aerosol missmatches with respect to the lookup table values (table 1). However our sensitivity tests which lead to the shown uncertainty bars in Figure 7 point to impacts below 10% uncertainty for most data points. The stability and size of the effect over a large number of flight situations and cloud geometries and the consistency with the expectation that non-vertical cloud surfaces would have just this effect, point to the 3D geometry as the reason for this result. Nonetheless, variation between cloud scenes (different colors) could also be due to albedo or aerosol variations. The method presented uses a constant offset for compensation. For the given ACRIDICON-CHUVA cases' typical observation perspectives and hence the dominant cloud surface orientation observed, this is found by comparison to stereo values. In the future, this limitation might be overcome by a method using "vertical cloud wall" results as presented as first guess, allowing for a first approximate derivation of cloud surface orientation and using forward simulations for varying cloud tilt to iteratively improve results. The computational effort necessary for such an approach exceeds the resources available for this study. Alternatively, a simple "calibration" of the method as presented could also be reached using a small set of selected 3D simulations. A small number a few hundred test results (as in our comparison to stereo points) at a low Monte Carlo accuracy - a high noise level - would be sufficient to characterise a single typical offset value. It could be provided at a limited computation effort.

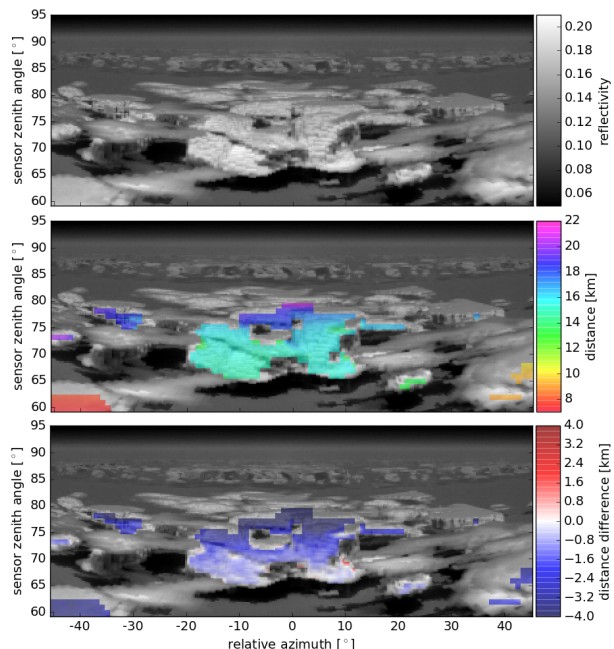

**Figure 8.** Demonstration of 3D impact on retrieval using an arbitrary cloud scene. (a) Based on ICON-LEM modelled cloud data the reflectivity for an observer at 7500 m height with the sun at $\theta_0 = 30°$ and the sun behind the observer ("to the south") and with a field-of-view between sensor zenith angle $60°$ and $95°$ and a relative sensor azimuth angle between $-45°$ and $+45°$ ("to the north") is simulated using the 3D radiative transfer model MYSTIC. (b) shows the true distance between observer and cloud for all cloud areas bright enough (reflectivity > 0.15) for elements of $1° \times 1°$. (c) shows the difference between these distances and the result of the presented oxygen-A derived values.

In Figure 8 an example of the 3D cloud geometry impact on the method is shown for a synthetic cloud observation. For this demonstration an available mid-latitude cumulus case was used. Of course this scene can not be expected to be fully comparable to the tropical cumulus congestus cases during the ACRIDICON-CHUVA campaign. Still it illustrates this possibility and at the same time displays once more a the impact of 3D cloud geometry at the observed order of magnitude. The case is taken from
5  an ICON-LEM (Icosyhedral Nonhydrostatic weather model - Large Eddy Model) run for Germany with a spatial resolution of $165 \times 165 \times 165$ m with a two-moment microphysics model package (Heinze et al., 2017). True cloud distances shown in Fig. 8b range between 8 and about 20 km. The application of the oxygen-A band retrieval to these simulations results in deviations of a few hundred metres to a few kilometres from the true distance. As already seen in the comparison of stereo and oxygen-A band values, for the synthetic test the deviation is negative almost erverywhere too (distance derived from oxygen-A is too
10  small). For this case the mean deviation over the whole scene is 2.1 km. As all parameters in this synthetic test case are fixed to the values used in the lookup-table of the method, only remaining candidate source of deviation is the specific 3D geometry of this cloud case. Here the central part of the cloud field systematically tilts away from the vertical with values close to vertical at the cloud lower edge and almost horizontal cloud top for the top line of analysed pixels.

In the future a consideration of such cloud surface orientation caused systematic differences could be possible by including additional forward solutions for arbitrarily tilted cloud surfaces using into the lookup-table with the help of far more extensive 3D radiative transfer simulations and iterative approaches. The overall accuracy of oxygen-A derived heights apart from this cloud geometry impact is comparable to the accuracy of a stereo image based approach for the application to aircraft observations, A clear advantage of the presented absorption method is that it is not limited to individual features of high image contrast (as the stereo method), but provides distance over large continuous areas (apart from cloud tops and shadows). It provides the cloud heights on the sensor coordinate grid which facilitates evaluation of other information on the same grid.

Eventually a full integration of stereo and absorption approach into one method might be a beneficial approach, as advantages of both methods could be combined. At cloud tops and horizontal edges where the oxygen absorption is strongly affected by 3D cloud geometry effects, stereo methods can most easily provide information as image contrast is high. In cloud regions where geometry variation is small, it is impossible to identify points of high contrast for the stereo method, but oxygen absorption still provides reliable information.

The actual specific goals of the presented retrieval are the provision of vertical location for the determination of particle size and phase profiles of convective cloud fields from spectral cloud side remote sensing and the provision of cloud surface orientation for improved microphysics retrievals (Jäkel et al., 2017; Ewald et al., 2018). To this end, oxygen-A band based derivation of this height information proofed to be an elegant approach and examples of use can be found in Polonik et al. (2018).

*Author contributions.* U. Schwarz developed a first tested the approach and sensitivities in a Master's thesis. Based on his work T. Zinner developed the presented method, carried out the presented radiative transfer simulations and data evaluation and wrote the manuscript. T. Koelling developed the software providing easy access to specMACS data and facilitated the automated evaluation of specMACS data. F. Ewald and T. Kölling were in charge of the measurements during the ACRIDICON-CHUVA campaign. E. Jaekel carried out the stereo analysis of the scenes used for the comparison of the two methods and revised the manuscript with focus on stereo methods. E. Jaekel, T. Koelling, F. Ewald, B. Mayer, M. Wendisch contributed to the final form of the manuscript text.

*Acknowledgements.* The ACRIDICON-CHUVA campaign was supported by the Max Planck Society (MPG), the German Science Foundation (DFG Priority Program SPP 1294), the German Aerospace Center (DLR), the FAPESP (Sao Paulo Research Foundation) grants 2009/15235-8 and 2013/05014-0, and a wide range of other institutional partners. It was carried out in collaboration with the USA–Brazilian atmosphere research project GoAmazon2014/5, including numerous institutional partners. We would like to thank Instituto Nacional de Pesquisas da Amazonia (INPA) for the local logistic help prior to, during, and after the campaign. Thanks also to the Brazilian Space Agency (AEB: Agencia Espacial Brasileira) responsible for the program of cooperation (CNPq license 00254/2013-9 of the Brazilian National Council for Scientific and Technological Development). The entire ACRIDICON-CHUVA project team is gratefully acknowledged for collaboration and support. We thank F. Jakub for providing the ICON-LEM data used in the discussion section. F. Ewald was supported by the German Research Foundation (DFG) under grant number MA 2548/9-1. T. Kölling was supported by the German Research Foundation (DFG) under grant number Zi 1132/3-1.

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
