# Peer review of "Cloud geometry from oxygen-A band observations through an aircraft side window"

_Atmospheric Measurement Techniques, 2018_

## Referee Comment (RC1) · Anonymous Referee #3 · 2 Oct 2018

The paper presents an algorithm for determination of cloud shape/geometry using cross-track scanning spectrometer making measurements in O2 A-band. The authors made an extensive retrieval error estimation effort and honestly discuss challenges facing the proposed approach. These challenges appear to be so serious that the whole retrieval algorithm needs substantial modification (see my comments below for details). I hope that the authors will be able to improve their technique during the discussion and reply to reviews period, so the paper could be then accepted after major changes.

General comments ——————-

The main issue affecting the proposed retrieval technique is multiple scattering of light in essentially 3D cloud geometry. This creates a kind of a circle: cloud shape is determined point-by-point using LUT based on 3D radiation transfer model, which itself has

to imply a specific shape of cloud as a whole. Simplified assumption of cloud shape ("cloud wall") leads to substantial bias (3.8 km) in the derived cloud position compared to an independent stereo dataset (which is assumed to be more robust).

The authors should also note that the same stereo dataset cannot be used for both correction of A-band retrievals and for validation of them. The 3.8-km offset applied to A-band data (making it to agree better with stereo results) is discussed in the paper, however, it is not even mentioned in the abstract when the accuracy of the proposed approach is described. This misleads the reader.

The paper in its present form describes a technique that currently does not work, and its results are artificially forced to agree with stereo dataset for "validation". This is certainly is not worth publishing. In my opinion the only way to save the paper is to follow the authors' own suggestion (last paragraph of Conclusions Sec.) and incorporate stereo measurements into A-band algorithm as an occasional "calibration" source. (Note that stereo measurements may not be used for validation then.) Only after this is successfully done (which is a "major change") paper can be accepted for publication.

English in the paper is generally acceptable except for few sentences that need clarification. However, the paper (Conclusions in particular) would certainly benefit from proof-reading by a native English speaker.

Line comments ————-

p.1, l.1: Is ACRIDICON-CHUVA an acronym? If yes, spell it out (may be in the paper text rather than in the abstract).

p.1, ll.12-14: The last sentence of abstract is misleading (see general comment above), since it does not mention the 3.8-km offset subtracted from A-band retrievals. This operation makes A-band retrievals dependent on stereo dataset, which then cannot be used as direct validation source.

p.1, ll.13-14: Distance accuracy should be compared with typical intrinsic property of

cloud field, such as horizontal scale (what was typical cloud size?), not with typical distance between cloud and aircraft.

p.3, l.14: Making measurements through aircraft window (rather than through instrument's own window) may present a problem. Any window can be dirty or scratched, and this should be factored into instrument's calibration. Unlike instrument's own window, aircraft's window cannot be taken into calibration lab. How specMACS was calibrated? Can additional absorption by the window potentially bias A-band measurements?

p.3. l.18: Add publication year to "Ewald et al."

p.6., ll.16-19: This paragraph needs clarification: has the spherical cloud been moved up and down relative to the sensor height in order to achieve different cloud surface angles?

p.7. Fig.7m: "sphere angle" is not defined or used in text.

p.7, caption; p.8, l.1; p.9, ll.11-12; p.10, l.6; etc.: Change "zenith" to "zenith angle".

p.7, caption: Use subscript 's' for sensor altitude 'z' (as it is used later in text).

p.7, caption: Caption is longer than the page, only part of it can be seen.

p.8, ll.7-19: Parts of this list are repeated in Table 1 and can be omitted.

p.8, l.8: The whole line is unclear. As viewing direction is not necessarily horizontal, what is "horizontal extent in viewing direction". What does "infinite, periodic perpendicular to it" mean? Please, clarify.

p.8, l.19: I suggest to include 'z_s' on this line, since it is used below it without definition.

p.8, ll.23-26: "For a given sensor height the cloud wall is visible under certain observation zenith angles only up to a certain distance." This implies certain finite cloud top and bottom heights, which are not defined or discussed. What are they in the model? This may also contradict l.8 (see above) stating that cloud is somehow "infinite". Please,

clarify.

p.9, l.14: The phrase "filtering cloud edge areas for the remaining cloud situations" is completely unclear. And what is "cloud edge" (also used in other places in the paper) compared to "cloud surface"?

p.10, l.4: I suggest to define "R = R_O2A" here as it is used without subscript or with different subscripts below this line.

p.10, eqs.2&3: I am confused. Dependences of R = R_O2A on all parameters presented in Fig. 3 seem to be monotonic. Then, R_min < R_LUT < R_max or R_max < R_LUT < R_min (depending on definition of R_min and R_max) for any parameter (pressure, etc.). In both cases R_LUT cancels from the numerator of Eq. (2), thus, Delta R = 2 sigma_R. Why this relation is not satisfied in Table 1? Why both Delta R and sigma_R are needed?

p.10, l.11: sigma_R defined by Eq. (3) should be called "relative dispersion" (ratio of the standard deviation to the mean) rather than "standard deviation".

p.11, Fig.5: Here and in the rest of the paper it is not clear what the authors mean by "distance": horizontal distance or distance along the viewing direction. Please, clarify.

p.11, Fig.5 and ll.3-17: Important: the absolute accuracies (in meters, not %) of distance measurement must be presented instead of relative ones. Measured distance is only an intermediate step towards derivation of cloud shape and geographic location, which are independent of the sensor position. Thus, only the absolute values of uncertainties in derived distance have physical meaning. The same, say, 600 m accuracy of cloud location determination (the only thing that matters) can be 1% if the distance between cloud and sensor is 60 km or 10% if this distance is 6 km, thus, speculation that 1% is better than 10% does not make any sense.

p.11, ll.3-5: The sentence sounds a bit awkward. I suggest: "At low altitudes most cloud sides can be observed looking upward, while for high altitudes they are observed

looking downward."

p.12, ll.15-16: "uncertainty of oxygen-A retrievals is likely to be larger than that of the stereo derivation": How much larger? Comparison is hard since the stereo accuracy is given in meters (200-300 m), while that of A-band measurements is given in Section 2.2.3 in % of the distance between cloud and sensor (see comment to Fig. 5 above).

p.12, l.23: How relative azimuth is defined in the paper? If here ∼180 deg azimuth means that the sun is behind the sensor, does it mean that the azimuth of 0 deg in Table 1 corresponds to the sun in front of the sensor (which would not make much sense)?

p.12, l.34: I suggest to replace "shown as well" by "also shown in Fig. 6".

p.13, Figs.6(c)&7(a): Are these distances horizontal or along sensor's line of view to the point?

p.13, Fig.6(c,d): It is difficult to see comparison between A-band and stereo distances/heights since the former are shown by color, while the latter - by numbers. I suggest to include (in addition to this figure) several single-scan plots showing 2D cross-section of cloud surface derived from A-band measurements with stereo points plotted over. This could also show how/if stereo points can be used for "calibration" of A-band retrievals on single scan or single cloud scene bases.

p.14, Fig.7: Make both plots square (since x and y have the same physical meaning and scale) and set them side by side. Provide means and standard deviations of (y-x).

p.14, Fig.7: The meaning of points' colors is not explained neither in text nor in caption. Do they correspond to the mentioned 27 cloud scenes?

p.14, ll.3-4: Replace "200 x 200 mˆ2" by "200 m x 200 m".

p.15, ll.9-10; p.16, ll.12-13: My impression from looking at Fig. 7(a) is that if different cloud scenes (assuming that they are identified by point colors) are considered separately the fitting offsets in them would significantly vary (and may even have different signs). I do not see convincing justification that all these scenes belong to the same "type" so the same 3.8-km offset is characteristic for all of them.

p.15, ll.18-19; p.16, ll.17-18: "In general, such a "calibration" of the method could also be reached using a limited number of synthetic cloud model based simulations." Here the authors try to downplay the complexity of 3D RT simulations contradicting their own words in the first paragraph of Sec. 2.2.2. I do not believe in "universal" 3D-RT LUT based on a few of cloud scenes.

p.16, l.9: Delete "towards the horizontal", change "shortcuts" to "decreases".

p.16, l.10: Replace "towards" by "for".

p.16. l.12: What is "simulated geometry path"?

p.16, ll. "Eventually a combination of stereo and absorption methods might be a beneficial approach, as advantages of both methods could be combined". This is what needs to be done in the next revision of this paper (not eventually), potentially creating a working retrieval method. Applying a campaign-wide offset to all A-band retrievals can be considered as a first approximation to a working synergistic approach in which stereo-based corrections should be made at each individual cloud scene.

---

## Author Comment (AC1) · 5 Oct 2018

Dear Reviewer 3,

Thank you for reviewing our manuscript. You provided your full review already during the quick access phase. That is why we already sent a full reply to the editor after the quick access reviews. To provide an improved manuscript to the other reviewers, we already did consider the changes to the manuscript text you requested before the discussion phase. That means, the other reviewers already see a manuscript version which includes the changes discussed below.

We considered many of your smaller points and do agree, but we do not agree with your main point of criticism. Please see below for our point-by-point reply.

Best regards,
Tobias Zinner (and co-authors)

**Reply to reviewer 3:**

Reviewer comments are highlighted in gray.

I include my complete review. It can be used when/if the paper is posted in AMTD.

The paper presents an algorithm for determination of cloud shape/geometry using cross-track scanning spectrometer making measurements in O2 A-band. The authors made an extensive retrieval error estimation effort and honestly discuss challenges facing the proposed approach. These challenges appear to be so serious that the whole retrieval algorithm needs substantial modification (see my comments below for details). I hope that the authors will be able to improve their technique during the discussion and reply to reviews period, so the paper could be then accepted after major changes.

General comments
* * *
The main issue affecting the proposed retrieval technique is multiple scattering of light in essentially 3D cloud geometry. This creates a kind of a circle: cloud shape is determined point-by-point using LUT based on 3D radiation transfer model, which itself has to imply a specific shape of cloud as a whole. Simplified assumption of cloud shape ("cloud wall") leads to substantial bias (3.8 km) in the derived cloud position compared to an independent stereo dataset (which is assumed to be more robust).

The authors should also note that the same stereo dataset cannot be used for both correction of A-band retrievals and for validation of them. The 3.8-km offset applied to A-band data (making it to agree better with stereo results) is discussed in the paper, however, it is not even mentioned in the abstract when the accuracy of the proposed approach is described. This misleads the reader.

The paper in its present form describes a technique that currently does not work, and its results are artificially forced to agree with stereo dataset for "validation". This is certainly is not worth publishing. In my opinion the only way to save the paper is to follow the authors' own suggestion (last paragraph of Conclusions Sec.) and incorporate stereo measurements into A-band algorithm as an occasional "calibration" source. (Note that stereo measurements may not be used for validation then.) Only after this is successfully done (which is a "major change") paper can be accepted for publication.

English in the paper is generally acceptable except for few sentences that need clarification. However, the paper (Conclusions in particular) would certainly benefit from proof-reading by a native English speaker.

We do agree with the reviewer that we should have made limitations of our method even more clear already in the abstract. We extended abstract and description accordingly. However we do not think that a fundamental extension of our method with additional "calibration" stereo data is necessary or would change the message of this paper. Already in the original manuscript we do suggest the combination of both data sources for our purpose and discuss the limitations.

Our main goal is a determination of cloud surface orientation and cloud points' vertical height for our cloud side view for a specific campaign data set (ACRIDICON-CHUVA). For a plausibility check we compare our data to stereo-points. We find an offset (3.8 km) which is most likely caused by 3D effects. Apart from this deviation, errors are small and within the expected and described ranges. As 3D effects are the most likely explanation for sign and size of the offset, we suggest to correct the offset to match the mean stereo distance.

On purpose, we did not use the word "validation" in the manuscript for this comparison of stereo and oxygen-A derived distance, because we do not consider the stereo distances to be free of uncertainty and even systematic effects (e.g. a window distortion effect).

For one of our main goals, the determination of cloud surface orientation for the correction of 3D effects of other retrievals, the mentioned offset does not introduce any error. For our second goal, the determination of a cloud height for specMACS derived products, we would add an additional (and problematic) bias, if our attribution of the offset to 3D effects is not correct. The second point has been clarified in the revised manuscript.

Nonetheless, we consider our proposed method and its detailed analysis to be careful and sufficiently complete to be instructive for the growing community using spectral solar remote sensing methods and by our group for application in successive studies. Any following paper on use of oxygen-A band distance derivation in complex cloud situations, whether our own group's application or any other author's, could be compared to our open and detailed analysis in this manuscript.

Apart from these argumentation, it has to be emphasized that all passive solar cloud remote sensing techniques suffer from systematic 3D effects due to real clouds' shape, well known by the community from the very beginning, but only incrementally quantified over time. Nonetheless a vast number of important and significant publications used these techniques, the good ones keeping in mind the limitations introduced even without proper quantification.

**Point-by-point reply**

Line comments
* * *
p.1, l.1:
Is ACRIDICON-CHUVA an acronym? If yes, spell it out (may be in the paper text rather than in the abstract).

Changed in the text.

p.1, ll.12-14:
The last sentence of abstract is misleading (see general comment above), since it does not mention the 3.8-km offset subtracted from A-band retrievals. This operation makes A-band retrievals dependent on stereo dataset, which then cannot be used as direct validation source.

It is mentioned in the abstract now.

p.1, ll.13-14:
Distance accuracy should be compared with typical intrinsic property of cloud field, such as horizontal scale (what was typical cloud size?), not with typical distance between cloud and aircraft.

The typical cloud type was added.

p.3, l.14:
Making measurements through aircraft window (rather than through instrument's own window) may present a problem. Any window can be dirty or scratched, and this should be factored into instrument's calibration. Unlike instrument's own window, aircraft's window cannot be taken into calibration lab. How specMACS was calibrated? Can additional absorption by the window potentially bias A-band measurements?

The specMACS calibration in Ewald et al 2016 includes the window.

p.3. l.18:
Add publication year to "Ewald et al."

Added.

p.6., ll.16-19:
This paragraph needs clarification: has the spherical cloud been moved up and down relative to the sensor height in order to achieve different cloud surface angles?

Correct. The text has been clarified there.

p.7. Fig.7m:
"sphere angle" is not defined or used in text.

Changed to "surface angle" as used and defined in the text.

p.7, caption; p.8, l.1; p.9, ll.11-12; p.10, l.6; etc.:
Change "zenith" to "zenith angle".

I did a "Search and replace" throughout the document.

p.7, caption:
Use subscript 's' for sensor altitude 'z' (as it is used later in text).

Done.

p.7, caption:
Caption is longer than the page, only part of it can be seen.

Corrected. Seems to be a problem of the discussion format.

p.8, ll.7-19:
Parts of this list are repeated in Table 1 and can be omitted.

That is correct. The list has been removed.

p.8, l.8:
The whole line is unclear. As viewing direction is not necessarily horizontal, what is "horizontal extent in viewing direction". What does "infinite, periodic perpendicular to it" mean? Please, clarify.

That is the the horizontal extent in cross-track direction using the aircraft coordinate system (away from an aircraft). It is not the along-track (flight direction) extent. In the model setup this second dimension is periodic. A 2 km thick cloud wall is used. I have added a sentence to explain the geometry.

p.8, l.19:
I suggest to include 'z_s' on this line, since it is used below it without definition.

No this is not "z_s". z_s is sensor height while the derived height is the "z_s – z_o2a". It now reads "In order to check the validity of distance and height measurements derived ...".

p.8, ll.23-26:
"For a given sensor height the cloud wall is visible under certain observation zenith angles only up to a certain distance." This implies certain finite cloud top and bottom heights, which are not defined or discussed. What are they in the model? This may also contradict l.8 (see above) stating that cloud is somehow "infinite". Please, clarify.

See answer to l.8 above.

p.9, l.14:
The phrase "filtering cloud edge areas for the remaining cloud situations" is completely unclear. And what is "cloud edge" (also used in other places in the paper) compared to "cloud surface"?

I tried to clarify. It became "cloud tops, which are always more or less horizontal". Other appearance of "edge" was changed accordingly.

p.10, l.4:
I suggest to define "R = R_O2A" here as it is used without subscript or with different subscripts below this line.

Added your suggestion.

p.10, eqs.2&3:
I am confused. Dependences of R = R_O2A on all parameters presented in Fig. 3 seem to be monotonic. Then, R_min < R_LUT < R_max or R_max < R_LUT < R_min (depending on definition of R_min and R_max) for any parameter (pressure, etc.). In both cases R_LUT cancels from the numerator of Eq. (2), thus,
Delta R = 2 sigma_R.
Why this relation is not satisfied in Table 1? Why both Delta R and sigma_R are needed?

You are right. Your question led to confusion on my side too. Originally this was meant to provide some complicated but didactically nice derivation. Due to your question I had to reconstruct why the numbers in table 1 for DeltaR and sigmaR did not show a perfect factor 2. Obviously numbers of a preliminary analysis using DeltaR and a final analysis using sigmaR both made it into the table. I should have noticed that before. I have removed the DeltaR definition and part of the explanations from the text and the column with the incomplete DeltaR values from the table.

p.10, l.11:
sigma_R defined by Eq. (3) should be called "relative dispersion" (ratio of the standard deviation to the mean) rather than "standard deviation".

I would prefer to stick to "standard deviation from the mean".

p.11, Fig.5:
Here and in the rest of the paper it is not clear what the authors mean by "distance": horizontal distance or distance along the viewing direction. Please, clarify.

Tried to clarify in two places of the retrieval chapter.

p.11, Fig.5 and ll.3-17:
Important: the absolute accuracies (in meters, not %) of distance measurement must be presented instead of relative ones. Measured distance is only an intermediate step towards derivation of cloud shape and geographic location, which are independent of the sensor position. Thus, only the absolute values of uncertainties in derived distance have physical meaning. The same, say, 600 m accuracy of cloud location determination (the only thing that matters) can be 1% if the distance between cloud and sensor is 60 km or 10% if this distance is 6 km, thus, speculation that 1% is better than 10% does not make any sense.

I would like to leave this point open for the public discussion. Use of relative values shows differences more nicely and are more intuitive to understand. Using absolute values would basically only show an increasing stddev with increasing distance. In comparison literature usually both are provided.

p.11, ll.3-5:
The sentence sounds a bit awkward. I suggest:
"At low altitudes most cloud sides can be observed looking upward, while for high altitudes they are observed looking downward."

I used your suggestion. Thank you.

p.12, ll.15-16:
"uncertainty of oxygen-A retrievals is likely to be larger than that of the stereo derivation": How much larger? Comparison is hard since the stereo accuracy is given in meters (200-300 m), while that of A-band measurements is given in Section 2.2.3 in % of the distance between cloud and sensor (see comment to Fig. 5 above).

I have tried to clarify that and to add a few numbers from the earlier publications. They usually provide only a few numbers, and usually absolute values together with typical cloud distance as well as relative numbers.

p.12, l.23:

How relative azimuth is defined in the paper? If here ~180 deg azimuth means that the sun is behind the sensor, does it mean that the azimuth of 0 deg in Table 1 corresponds to the sun in front of the sensor (which would not make much sense)?

Thanks for pointing out this inconsistency. I have corrected the values in table 1.

p.12, l.34:
I suggest to replace "shown as well" by "also shown in Fig. 6".

Done.

p.13, Figs.6(c)&7(a):
Are these distances horizontal or along sensor's line of view to the point?

See above.

p.13, Fig.6(c,d):
It is difficult to see comparison between A-band and stereo distances/heights since the former are shown by color, while the latter - by numbers. I suggest to include (in addition to this figure) several single-scan plots showing 2D cross-section of cloud surface derived from A-band measurements with stereo points plotted over. This could also show how/if stereo points can be used for "calibration" of A-band retrievals on single scan or single cloud scene bases.

You mean a 2D cross section in x-z plane? x being the horizontal direction along line-of-sight, across-track. That would mean to show one x-z cross-section through the oxygen-A derived surface as a line for each stereo value (sometimes two), because the surface changes quickly in y (flight) direction. We are not sure, if another figure improves this comparison much. A third chance to directly compare them is given in Fig 7 where the turquoise points show the values again.

p.14, Fig.7:
Make both plots square (since x and y have the same physical meaning and scale) and set them side by side. Provide means and standard deviations of (y-x).

Will be done.

p.14, Fig.7:
The meaning of points' colors is not explained neither in text nor in caption. Do they correspond to the mentioned 27 cloud scenes?

Yes. Included in text and caption.

p.14, ll.3-4:
Replace "200 x 200 m^2" by "200 m x 200 m".

Done.

p.15, ll.9-10; p.16, ll.12-13:
My impression from looking at Fig. 7(a) is that if different cloud scenes (assuming that they are identified by point colors) are considered separately the fitting offsets in them would significantly vary (and may even have different signs). I do not see convincing justification that all these scenes belong to the same "type" so the same 3.8-km offset is characteristic for all of them.

That there is variation is certainly true and to be expected as not two clouds will display the exact same geometry and 3D effects. On the other hand not each single stereo result is to be trusted 100%. For all larger groups the offset lies between 0 and 6 km. Only a clear minority shows the opposite sign and most of them show large uncertainty bars labeling them as difficult situations. The use of a single offset for the whole campaign data set is certainly a compromise limiting accuracy. On the other hand, we are confident that our overall uncertainty values are correct, as they are nicely confirmed by this comparison (after offset correction). Use of one offset of 3.8 km might of course leave individual cloud scenes with systematic remaining biases of mostly 1-2 km. We mention this in the text now.

p.15, ll.18-19; p.16, ll.17-18:
"In general, such a "calibration" of the method could also be reached using a limited number of synthetic cloud model based simulations." Here the authors try to downplay the complexity of 3D RT simulations contradicting their own words in the first paragraph of Sec. 2.2.2. I do not believe in "universal" 3D-RT LUT based on a few of cloud scenes.

We tried to clarify.

"In general, such a ``calibration`` of the method could also be reached using a small set of selected 3D simulations. A small number a few hundred test results (as in our comparison to stereo points) at a low Monte Carlo accuracy - a high noise level - would be sufficient to characterise a single typical offset value. It could be provided at a small computation effort (several orders of magnitude below the effort to provide non-vertical slopes). "

For explanation: A use of a limited number of cloud resolving model produced scenes can be very effectively probed by a limited number of randomly selected single line-of-sight simulations, say 300 as in our stereo-comparison. Monte Carlo accuracy can be quite low for these 300 as we only try to characterise one number – the offset. Compared to such a set of simulations, even the "cloud wall" LUT simulations of the manuscript (4 SZA x 29 VZA x 17 distances x 10 altitudes = 19720) are still about 500 to 1000 more costly.

p.16, l.9:
Delete "towards the horizontal", change "shortcuts" to "decreases".

Done.

p.16, l.10:
Replace "towards" by "for".

I have replaced "towards" by "approaching".

p.16. l.12:
What is "simulated geometry path"?

Bad wording. Changed to "simulated absorption path".

p.16, ll.
"Eventually a combination of stereo and absorption methods might be a beneficial approach, as advantages of both methods could be combined". This is what needs to be done in the next revision of this paper (not eventually), potentially creating a working retrieval method. Applying a campaign-wide offset to all A-band retrievals can be considered as a first approximation to a

working synergistic approach in which stereo-based corrections should be made at each individual cloud scene.

Basically this is what we already do in this paper!? We use an oxygen-A retrieval to provide cloud distances for all cloud surfaces with uncharacterised 3D cloud effect. We use a very limited number of stereo derived points to characterise the campaign-typical 3D effect. We suggest to use the offset corrected oxygen-A results as we are confident that we understood all limitations after careful analysis. We do  not see how the message or validity of results of this manuscript would change, if we would use only half of the stereo points for "calibration" and use the other half for "validation".

---

## Referee Comment (RC2) · Anonymous Referee #1 · 11 Oct 2018

The paper presents a method to characterize the distance and height between airborne and cloud properties. The authors adapt an old concept based on the O2 A-Band absorption, usually applied to retrieve cloud top altitude of plane-parallel homogeneous cloud from satellite. The novelty and interest of the paper lie in the necessity to apply algorithm to finite clouds with sides. In this framework, authors have to realize extensive 3D radiative transfer simulations to develop look-up table based a "cloud wall". Several sensitivity tests were made concerning different geometry setup, cloud properties and cloud environment (aerosol, surface). At the end, comparisons of the distance retrieval with stereo measurement show a bias of 3.8 km that the authors attribute quite easily to 3D radiative effects.

At this point, I'm completely agree with the referee 3 that the paper cannot be published

without a validation of this assumption. Before accepting the paper, I request that the authors realize the simulation mentioned in page 15 and in the conclusions using cloud resolving model output and 3D radiative transfer simulation. Applying the algorithm to this simulated data will enable to confirm that 3D effects shortened the retrieved distance of an order of 3-4 kilometers and will strengthen the interest of the paper.

Other comments and questions:

1- In the introduction, the authors cite the papers demonstrating the concept of using O2 band to retrieve cloud top altitude (Yamamoto and Wark 1961, Wu, 1985, Fisher et Grass 1991) but do not mention the most recent papers as nothing were done since 1991. Can the authors actualize the references adding more recent bibliography concerning cloud top retrieval using O2-band?

2- Page 1, line 24, add reference for the retrieval of cloud top from brightness temperature.

3- Figure 1. For more clarity concerning the angle definitions, the authors should add the sensor zenith angle limits that are used for the sensitivities test and LUT computation.

4- Section 2.2.1. Present here the basic cloud used for the sensitivity test. How is the LWC and microphysics variability inside the cloud? Horizontally and vertically homogeneous?

5- Page 5, line 17: How to be sure you are in the saturation regime? If not what happens, is the distance shortened or stretched out?

6- Figure 3 and page 5, line 26: I found very weird and confusing to normalize the radiances to the minimum value. Normalizing them to the maximum value would allow to understand more easily the figure and the absorption differences according to the parameters.

7- Page 6, line 30-35. How to know if the cloud side is sufficiently vertical to apply the

AMTD
method?

8- Figure 4. Similar to suggestion 3. Explain clearly or with a schematic why with an airborne at zs=6km, the angular range of sensor zenith angle is between 71 and 99°.

9- Figure 5. Can you add or indicate the relative standard deviation value in percent?

10- Page 14. Line 3. Please begin to describe the figure 7 and how are select the grey dot before analyzing the figure.

11- Page 14- line 6: What is an "objective" analysis? How do you select the horizontal cloud deck points?

---

## Referee Comment (RC3) · Anonymous Referee #2 · 8 Nov 2018

This study examines the reliability of a promising method for estimating the distance and altitude of clouds observed by airborne radiometers. The analysis includes both a theoretical sensitivity study and the validation of results using independent measurements. I believe that the study presents significant results that will be of interest to the community and are worthy of publication. The methodology is sound and the presentation is clear. Even so, I recommend some important revisions before publication, mainly in explaining or discussing some key details. Please find my specific comments below.

Major:

Page 8, Lines 6-9: I am not sure if simulations would be more difficult and time consuming for tilted cloud sides than for vertical cloud sides. If one used the maximum cross

section method, the higher spatial resolution (required for tilted cloud sides) should not affect the computational demands as long as the volume extinction coefficients are in a similar range for tilted cloud sides as they were for vertical cloud sides.

Page 16, Lines 16-18: Does the stability of the 3.8 km offset mean that the tilt of cloud sides is similar in all observed scenes? I wonder why the observed scenes display less variability in the tilting of cloud sides than in aerosol properties or surface albedo (which were mentioned in Lines 15-16 as possible alternative explanations for the 3.8 km offset). Regarding Lines 19-23, I wonder if the assumption of tilted (and not vertical) cloud sides may work better in building look-up tables for future studies, as this could reduce or even eliminate the required offsets. Finally, it would help to mention whether and how the offset (or the typical cloud side tilt) may be obtained in future cases where stereo (or lidar or radar) data is not available.

Minor:

Page 4, Line 16: I suggest adding "As mentioned above, " (in Lines 8-9) to the beginning of the sentence "Figure 2 shows the spectral region of the oxygen-A band at different spectral resolutions. ".

Page 4 last line: "Detail like the" should be replaced by "Details like the".

Page 6, Lines 27-30: It would help to point out that the spherical cloud is shifted only up and down but not sideways.

Page 8, lines 3-4: The sentence "As mentioned before, the geometry of the observation situation involves time consuming Monte Carlo simulations to simulate these ratios." should be reworded for improved clarity. I also suggest refining the subsequent sentence, for example by adding a comma after "consequently".

Page 8, Line 30: What exactly is meant by 1.5% in the sentence "Here the standard deviation is 1.5%."? Is it a relative or absolute quantity, and is it for reflectance or A-band ratio?

Page 8, Lines 32-33: "Sensor zenith angle" is mentioned twice. I guess one of them should be switched to "solar zenith angle".

Page 12: It would help to include a brief discussion of stereo distance uncertainties-for example the uncertainties due to variations in camera pointing direction (caused by slight changes in aircraft attitude)-or at least to mention that uncertainties are discussed in Jakel et al. (2017).

Page 14, Line 5-8: It would be important to mention how the 233 excluded points were identified. Did subjective manual analysis or quantitative criteria decide whether a point was too close to (or even part of) a cloud top, or occurred in a shadow?

Page 15, Lines 9-10: It would help to discuss here why 3D effects can reduce the Oxygen A-band distances. This is discussed in detail in the conclusion section (Page 16 Lines 11-18) but I feel that the discussion would fit better into the main text than the conclusions. Also, it would help to explain or illustrate (either here or in Page 6) why photon pathlengths are shorter for tilted cloud sides than for vertical cloud sides.

Page 15, Line 32: the word "band" seems to be missing between "A" and "absorption". In the subsequent line, "straightforward" should be a single word.

Page 16, Line 2: The word "is" is missing between "goal" and "the".

Page 17, Line 2: "Proofed" should be replaced by "proved".

---

## Author Comment (AC2) · 7 Jan 2019

Please see supplement PDF.

Please also note the supplement to this comment:
https://www.atmos-meas-tech-discuss.net/amt-2018-220/amt-2018-220-AC2-supplement.pdf

---

## Author Comment (AC4) · 7 Jan 2019

Dear Reviewer 1,

thank you for your review. We now included a new analysis as reaction to your and other reviewers requests: A synthetic cloud test field from a cloud resolving model and a simulation of measurements with the 3D radiative transfer code demonstrate how O2A derived distances could be "calibrated" for certain cloud types as long as the type of cloud geometry expected can be provided by cloud modelling. The results largely corroborate our earlier conclusions.

Please find below our reply to your points.

Best regards,
Tobias Zinner (and co-authors)

**Reply to reviewer 1**

Reviewer comments are highlighted in gray.

The paper presents a method to characterize the distance and height between airborne and cloud properties. The authors adapt an old concept based on the O2 A-Band absorption, usually applied to retrieve cloud top altitude of plane-parallel homogeneous cloud from satellite. The novelty and interest of the paper lie in the necessity to apply algorithm to finite clouds with sides. In this framework, authors have to realize extensive 3D radiative transfer simulations to develop look-up table based a "cloud wall". Several sensitivity tests were made concerning different geometry setup, cloud properties and cloud environment (aerosol, surface). At the end, comparisons of the distance retrieval with stereo measurement show a bias of 3.8 km that the authors attribute quite easily to 3D radiative effects.
At this point, I'm completely agree with the referee 3 that the paper cannot be published without a validation of this assumption. Before accepting the paper, I request that the authors realize the simulation mentioned in page 15 and in the conclusions using cloud resolving model output and 3D radiative transfer simulation. Applying the algorithm to this simulated data will enable to confirm that 3D effects shortened the retrieved distance of an order of 3-4 kilometers and will strengthen the interest of the paper.

We hope that we accounted for your concerns by the new synthetic demonstration case added to the manuscript. It serves as a demonstration how stereo data could be replaced by a statistically generated set of Monte Carlo simulations for modeled cloud geometry with given typical computational capabilities. In addition, we think that the manuscript lays out the way for the community to minimize this approach's remaining uncertainties using future increased computational capabilities.

Our main goal is a determination of cloud surface orientation and cloud points' vertical height for our cloud side view for a specific campaign data set (ACRIDICON-CHUVA). For a plausibility check we compare our data to stereo-points. We find an offset (3.8 km) which is mainly caused by 3D effects. Apart from this deviation, errors are small and within the expected and described ranges. We now demonstrate, that the offset found lies in the range of offsets caused solely by the typical deviation of 3D cloud surface orientations.

Other comments and questions:

1- In the introduction, the authors cite the papers demonstrating the concept of using O2 band to retrieve cloud top altitude (Yamamoto and Wark 1961, Wu, 1985, Fisher et Grass 1991) but do not mention the most recent papers as nothing were done since 1991. Can the authors actualize the references adding more recent bibliography concerning cloud top retrieval using O2-band?

We added some most recent activities in the field.

2- Page 1, line 24, add reference for the retrieval of cloud top from brightness temperature.

Added.

3- Figure 1. For more clarity concerning the angle definitions, the authors should add the sensor zenith angle limits that are used for the sensitivities test and LUT computation.

I'm not sure, if I get you right here. Figure 1 is just a general illustration. We extended the angle information a bit.

4- Section 2.2.1. Present here the basic cloud used for the sensitivity test. How is the LWC and microphysics variability inside the cloud? Horizontally and vertically homogeneous?

Information was given with Figure 3. Added it here.

5- Page 5, line 17: How to be sure you are in the saturation regime? If not what happens, is the distance shortened or stretched out?

We test that during the sensitivity tests Figure 3d and 3k. Nothing changes for increasing line of sight optical thickness.

6- Figure 3 and page 5, line 26: I found very weird and confusing to normalize the radiances to the minimum value. Normalizing them to the maximum value would allow to understand more easily the figure and the absorption differences according to the parameters.

We think that this is rather a matter of taste. We had the same discussion in the author group from time to time, but did not reach and mutual agreement. This is the way it was defined in some of the basic literature and we decided to stick with it.

7- Page 6, line 30-35. How to know if the cloud side is sufficiently vertical to apply the method?

A considerable part of the later analysis and discussion in the paper is now spend on this topic.

8- Figure 4. Similar to suggestion 3. Explain clearly or with a schematic why with an airborne at zs=6km, the angular range of sensor zenith angle is between 71 and 99◦ .

Added comments to the caption.

9- Figure 5. Can you add or indicate the relative standard deviation value in percent?

I do not understand? The figure shows the standard deviations in percent.

10- Page 14. Line 3. Please begin to describe the figure 7 and how are select the grey dot before analyzing the figure.

We shifted around the sentences in this paragraph and moved the information about the grey point towards the beginning. I hope it's more clear now.

11- Page 14- line 6: What is an "objective" analysis? How do you select the horizontal cloud deck points?

The are identified using the retrieved oxygen-A distances. While their absolute position most likely carries a large error, the position of cloud deck point relative to each other is correct: the height of these areas does not show variation → a horizontal cloud deck.

---

## Author Comment (AC3)

Dear Reviewer 2,

thank you for your review. In the new revision we tried to consider your comments and we included a new analysis as reaction to the other reviewers requests: A synthetic cloud test field from a cloud resolving model and a simulation of measurements with the 3D radiative transfer code demonstrate how O2A derived distances could be "calibrated" for certain cloud types as long as the type of cloud geometry expected can be provided by cloud modelling. The results largely corroborate our earlier conclusions.

Please find below our reply to your review.

Best regards,
Tobias Zinner (and co-authors)

**Reply to reviewer 2**

Reviewer comments are highlighted in gray.

This study examines the reliability of a promising method for estimating the distance and altitude of clouds observed by airborne radiometers. The analysis includes both a theoretical sensitivity study and the validation of results using independent measurements. I believe that the study presents significant results that will be of interest to the community and are worthy of publication. The methodology is sound and the presentation is clear. Even so, I recommend some important revisions before publication, mainly in explaining or discussing some key details. Please find my specific comments below.

Major:

Page 8, Lines 6-9: I am not sure if simulations would be more difficult and time consuming for tilted cloud sides than for vertical cloud sides. If one used the maximum cross section method, the higher spatial resolution (required for tilted cloud sides) should not affect the computational demands as long as the volume extinction coefficients are in a similar range for tilted cloud sides as they were for vertical cloud sides.

The reason for the high computational demand of titled cloud sides compared to vertical ones lies in the large number of grid cells that have to be set up in the simulation domain. For a 10 km distance and a vertical cloud of 2 km horizontal extent, only 12 1km-sized grid boxes are needed. To simulate a smooth tilted slope, 50 m resolution is needed extending the domain to 240 50m-sized grid boxes. For the Monte Carlo method this increases the time for tracing photons through this grid by about factor 200. We included some new comments explaining this in the manuscript.

Page 16, Lines 16-18: Does the stability of the 3.8 km offset mean that the tilt of cloud sides is similar in all observed scenes? I wonder why the observed scenes display less variability in the tilting of cloud sides than in aerosol properties or surface albedo (which were mentioned in Lines 15-16 as possible alternative explanations for the 3.8 km offset).

Yes, it implies that we observe a predominant type of cloud orientation. We do not expect aerosol or surface albedo to cause variation within the single analysed scenes of around a minute/ a few 10s of

km distance. Nonetheless, the typical aerosol/albedo situation could affect the scene's offset, but only to a secondary degree. We added some lines of discussion of this at the end of the manuscript.

Regarding Lines 19-23, I wonder if the assumption of tilted (and not vertical) cloud sides may work better in building look-up tables for future studies, as this could reduce or even eliminate the required offsets. Finally, it would help to mention whether and how the offset (or the typical cloud side tilt) may be obtained in future cases where stereo (or lidar or radar) data is not available.

We mention this additional possibility in the discussion now.

Minor:

Page 4, Line 16: I suggest adding "As mentioned above, " (in Lines 8-9) to the beginning of the sentence "Figure 2 shows the spectral region of the oxygen-A band at different spectral resolutions. ".

Done

Page 4 last line: "Detail like the" should be replaced by "Details like the".

Done

Page 6, Lines 27-30: It would help to point out that the spherical cloud is shifted only up and down but not sideways.

Mentioned now.

Page 8, lines 3-4: The sentence "As mentioned before, the geometry of the observation situation involves time consuming Monte Carlo simulations to simulate these ratios." should be reworded for improved clarity. I also suggest refining the subsequent sentence, for example by adding a comma after "consequently".

Done.

Page 8, Line 30: What exactly is meant by 1.5% in the sentence "Here the standard deviation is 1.5%."? Is it a relative or absolute quantity, and is it for reflectance or A-band ratio?

Added "of reflectivity results".

Page 8, Lines 32-33: "Sensor zenith angle" is mentioned twice. I guess one of them should be switched to "solar zenith angle".

Thanks. Corrected.

Page 12: It would help to include a brief discussion of stereo distance uncertainties-for example the uncertainties due to variations in camera pointing direction (caused by slight changes in aircraft attitude)-or at least to mention that uncertainties are discussed in Jakel et al. (2017).

We mention reasons for general stereo uncertainty now.

Page 14, Line 5-8: It would be important to mention how the 233 excluded points were identified. Did subjective manual analysis or quantitative criteria decide whether a point was too close to (or even part of) a cloud top, or occurred in a shadow?

It was a subjective analysis. We removed the missleading "objective". A fully automated shadow and cloud masking is a topic for a whole series of further studies. We did it manually.

Page 15, Lines 9-10: It would help to discuss here why 3D effects can reduce the Oxygen A-band distances. This is discussed in detail in the conclusion section (Page 16 Lines 11-18) but I feel that the discussion would fit better into the main text than the conclusions. Also, it would help to explain or illustrate (either here or in Page 6) why photon pathlengths are shorter for tilted cloud sides than for vertical cloud sides.

This is not easy to do and understand. We had the feeling that such a lengthy discussion here would distract from the line of argumentation.

Page 15, Line 32: the word "band" seems to be missing between "A" and "absorption". In the subsequent line, "straightforward" should be a single word.

Done.

Page 16, Line 2: The word "is" is missing between "goal" and "the".

Done.

Page 17, Line 2: "Proofed" should be replaced by "proved".

Done.